# Optical Properties of BaAl_2_O_4_ Due to Cerium Doping and Heat Treatment in Different Atmospheres

**DOI:** 10.3390/mi16060688

**Published:** 2025-06-07

**Authors:** Montserrat Nevai Coyotl Ojeda, Benito de Celis Alonso, José Eduardo Espinosa Rosales, Epifanio Cruz-Zaragoza, Martín Rodolfo Palomino Merino

**Affiliations:** 1Facultad de Ciencias Físico Matemáticas, Benemérita Universidad Autónoma de Puebla, (BUAP), Avenida San Claudio y 18 Sur, Colonia San Manuel, Ciudad Universitaria, Puebla P.C. 72570, Mexico; montserrat.coyotlo@alumno.buap.mx (M.N.C.O.); bdca@fcfm.buap.mx (B.d.C.A.); espinosa@fcfm.buap.mx (J.E.E.R.); 2Instituto de Ciencias Nucleares, Universidad Nacional Autónoma de México, A.P. 70-543, Ciudad de México P.C. 04510, Mexico; ecruz@nucleares.unam.mx

**Keywords:** barium aluminate, cerium dopant, photoluminescence, air atmosphere, nitrogen-reduced atmosphere, optical properties

## Abstract

The luminescent properties of cerium-doped barium aluminate (BaAl_2_O_4_) samples with varying Ce concentrations (0–1.1 mol%) prepared either in an air or nitrogen-reduced atmosphere are presented. This work provides the first detailed comparison of the material’s structural, luminescent, and chromatic properties at different doping levels and thermal treatments. X-ray diffraction analysis confirmed the hexagonal crystal structure of barium aluminate. Samples treated in an air atmosphere exhibited crystallite sizes of 58.5 nm for undoped samples and 45.7 nm for doped samples. In contrast, those treated under nitrogen showed smaller crystallite sizes, i.e., 39.8 nm for undoped and 42.3 nm for doped samples, respectively. XPS analysis indicated that the nitrogen-reduced atmosphere minimized Ce oxidation, favoring the presence of Ce^3+^. The bandgap values of the material were 4.0 eV and 5.6 eV for the air and for the nitrogen atmosphere, respectively. Photoluminescence spectra showed maxima at 357 nm (air) and 386 nm (nitrogen), attributed to *4f-5d* transitions of Ce. The samples under air atmosphere showed longer lifetimes values (0.94 ns) compared to those in a nitrogen atmosphere (0.40 ns). These results suggest that thermal treatment in an air atmosphere promoted better structural order and higher photoluminescence efficiency, while treatment in a nitrogen-reduced atmosphere increased defect formation, shortening the lifetime. Chromaticity coordinate analysis showed that the cerium ion dopant influenced the blueish emission color in both samples.

## 1. Introduction

Barium aluminate (BaAl_2_O_4_) is an important class of phosphorescent inorganic material because of its high melting point, high thermal and chemical stability, low density, catalytic activity, radiation resistance, and long-afterglow phosphorescence [1,2,3]. The electronic structure of these materials can be modified by introducing dopants during manufacturing and by varying the synthesis method [4]. Dopants intentionally added within a matrix influence the electronic properties of the material by altering the density of charge carriers within the bandgap and the energy levels of the electronic states as well as the carrier mobility. The synthesis methods affect the electronic material structure, chemical composition, and defect distribution. Research shows that the physicochemical properties of the materials, as well as their persistent emission, are associated with their synthesis route and the concentration of the dopant [5,6]. Phosphorescent materials find applications in emergency signals, radiation detection, X-ray imaging, LEDs, etc. [3,7,8].

It is well known that barium aluminate phosphors doped with various rare earth elements, such as Eu^2+^, Tb^3+^, Ce^3+^, Eu^3+^, Er^3+^, Dy^3+^, Tm^3+^, Tb^3+^, and Pr^3+^*,* exhibit enhanced luminescence [1,7,9,10,11,12]. The long-afterglow phosphorescence of BaAl_2_O_4_ was analyzed [13], which exhibited a duration longer than 10 h at 402 and 450 nm [13]. The spectrum showed a broad emission band over 300–650 nm, peaking at 405 nm. This band was attributed to defect state transitions in the BaAl_2_O_4_ matrix. Gedekar et al. [8] observed PL emissions of ultraviolet light from cerium-doped barium aluminate at 386 nm, attributed to 5d→4f electronic transitions of Ce^3+^. These emissions depended on the host matrix and varied from the UV region to the visible region. Strongly blue-emitting Ce^3+^-activated BaAl_2_O_4_ nanophosphors with an emission peak at 422 nm were observed [7]. In 2024, Hema et al. [13] documented yellow photoluminescent emissions at 589 nm from BaAl_2_O_4_: Ce^3+^.

Investigations of thermal treatment at high temperatures and of environmental atmospheres have shown they affect the internal structure of the material [14]. Trivalent cerium (Ce^3+^) presents poor thermal stability and fast oxidation during heat treatment above 300 °C. It exhibits redox properties and oxygen storage capacity due to its reversible transformation between the Ce^4+^ and Ce^3+^ states [15,16,17]. It is important to determine whether cerium is in its trivalent or tetravalent state when incorporated into a host matrix, as the atmosphere used during the material synthesis process can significantly influence the state of cerium oxidation. The use of an atmosphere over another can promote the appearance of either Ce^3+^ or Ce^4+^, thus altering the material properties. Understanding how these variables affect the material behavior is crucial to optimize its luminescent properties.

In this article, the influence of the heat treatment preparation and atmosphere conditions under which BaAl_2_O_4_: Ce samples were synthesized was assessed through the study of their crystalline structure, composition, and optical properties. Although the material is well known, this work presents the first detailed experimental comparison of its structural, luminescent, and chromatic properties at various doping concentrations, depending on the thermal treatment in air or nitrogen.

## 2. Experimental Section

### 2.1. Preparation of BaAl_2_O_4_: Ce Samples

The precursors and reagents used for the synthesis of each sample were aluminum isopropoxide (C_9_H_21_AlO_3_; Sigma-Aldrich, Saint Louis, MO, USA), barium nitrate (Ba(NO_3_)_2_; E. Merck AG, Darmstadt, Alemania), cerium III nitrate (Sigma-Aldrich), ethyl alcohol (C_2_H_5_OH; J.T. Baker, Phillipsburg, NJ, USA), acetic acid (C_2_H_4_O_2_; J. T. Baker), and distilled water. Aluminum isopropoxide and barium nitrate were mixed using an agate mortar, then dissolved in ethanol. The solution was then placed on a magnetic stirrer for 90 min at 500 rpm to obtain a homogeneous solution. Acetic acid and cerium dissolved in distilled water were then added to the first solution and stirred until the solution turned into a gel. To remove the remaining solvent, the gel was dried in a muffle furnace (KSL-1100X, Xian Yima Optoelec Co., Ltd., Shaanxi, China) at 120 °C for 120 min. A course powder was obtained and ground using an agate mortar until a finer powder was obtained and then treated with heat at 900 °C for 4 h. Finally, the obtained powder was further ground to achieve nanometric grain sizes for characterization.

Two series of BaAl_2_O_4_ samples doped with 0, 0.3, 0.5, 0.7, 0.9, and 1.1 mol% Ce were prepared. The first series, labeled Air Atm, was heat-treated at 900 °C for 4 h in air, whereas the second series, labeled N_2_ Atm, underwent the same heat treatment in a reduced atmosphere, with nitrogen used as the reducing agent.

### 2.2. Characterization of BaAl_2_O_4_: Ce Samples

The crystalline phase of the BaAl_2_O_4_ samples was identified by X-ray diffraction (XRD) using BRUKER diffraction equipment and a Cu Kα radiation source (λ = 1.5460 Å), operated at 40 kV and 40 mA for 2θ values in a range from 5° to 90° with increments of 0.04°. The chemical state of the elements was analyzed by X-ray photoelectron spectroscopy (XPS). These data were obtained using an XR50 M monochromatic Al Kα (hν = 1486.7 eV) X-ray source and a Phoibos 150 spectrometer with the one-dimensional detector 1D-DLD provided by SPECS (Berlin, Germany). Samples were mounted on a steel sample holder using double-sided copper tape. The peak-fitting analysis was performed by employing the AAnalyzer software, version 1.5. The bandgap of the material was determined using diffuse reflectance patterns obtained with a Varian Cary 5000 UV-Vis-NIR spectrophotometer equipped with an integrating sphere, covering a spectral range of 200–800 nm. The PL properties were analyzed by PL spectroscopy using the Horiba Jobin Yvon spectrofluorometer model NanoLog equipped with a 450 W Xenon lamp. Emission spectra were taken in the 350–600 nm range with an excitation wavelength of 300 nm. Fluorescence lifetime measurements were carried out using the same system.

## 3. Results and Discussion

Figure 1 shows the XRD patterns for undoped and *Ce*-doped BaAl_2_O_4_ samples heat-treated in an oxidizing atmosphere (Air Atm), while Figure 2 displays the same results for the samples obtained in a reduced atmosphere (N_2_ Atm). In Figure 1c, the crystal structure of BaAl_2_O_4_ is presented, corresponding to the space group P6_3_22, which exhibits a hexagonal crystal structure. The lattice parameters for each sample are listed in Table 1. A peak around 24° was observed in all XRD spectra, indicating the presence of a secondary phase identified as barium carbonate (BaCO_3_). This impurity was likely due to the reaction between Ba^2+^ and CO_2_ during the synthesis process, especially under the influence of residual moisture, and is not attributed to the addition of cerium. The formation of BaCO_3_ in barium-based aluminates has been reported in previous works [3,18].

Both groups of samples showed diffraction patterns closely matching the standard JCPDS card No. 00-017-0306 (Figure 1a and Figure 2a). Hema et al. [18] reported on Ce-doped BaAl_2_O_4_ samples which were prepared with the solid-state reaction method, with results similar to the ones presented here, after their samples were calcined twice: first at 1200 °C in an air atmosphere and then at 1100 °C with a surrounding reducing agent. In that work, BaCO_3_ was also observed as a secondary phase, suggesting that its presence was more related to synthesis conditions rather than cerium doping. The average crystallite size was calculated using the Scherrer equation [19], D=κλ/βcosθ, where D is the crystallite size, κ=0.9 is a constant, β is the full width at half maximum (FWHM) of the main peak in radians, and θ is the Bragg angle. This calculation was carried out using the peak of highest intensity, found around 28.4° for each sample. A magnified view of this maximum is shown in Figure 1b and Figure 2b.

For Air Atm samples, the average crystallite size was 58.5 nm for undoped samples and ranged between 44 and 46.9 nm for doped samples. In contrast, N_2_ Atm samples exhibited sizes of 39.8 nm for undoped samples and 41.2–44.7 nm for doped ones. These results suggest that the oxidizing atmosphere promoted greater crystallinity. The decrease in crystallite size with Ce addition in Air Atm samples may be attributed to dopant-induced lattice strain that inhibited grain growth [20]. Conversely, under reducing conditions, doping slightly increased crystallite size, possibly due to differences in defect structures or interactions with oxygen vacancies.

Figure 3a,b show the XPS spectra of BaAl_2_O_4_: Ce samples with a 0.5 mol% Ce concentration calcined in air or a nitrogen-reduced atmosphere. The peaks corresponding to *Al 2p*, *C 1s*, *O 1s*, *Ba 3d*, and *Ce 3d* can be observed in both spectra. In both cases, the peak corresponding to the *Ce 3d_3/2_* component overlap the Auger peak of Ba MNN [21], which could affect the interpretation of the *Ce 3d* peak.

For the same samples (0.5 mol% Ce), a deconvolution of the obtained curve attributed to Ce *3d* was performed; Figure 4a shows the curve for the air atmosphere. The peak corresponding to *p0* is thought to be a component of *Ba 3d*, specifically the Auger component MNN. Peak *p1* is attributed to the Ce^4+^ oxidation state, and peak *p2* is attributed to Ce^3+^, as has been reported previously [22]. Similarly, Figure 4b shows the deconvolution for the N_2_ Atm sample, performed using the same fitting parameters. In this spectrum, it can be observed that the preserved oxide corresponds to the Ce^3+^ oxidation state, while the *p1* peak corresponding to Ce^4+^ is almost absent.

The determination of the band gap energy E_g_ was carried out using diffuse reflectance data and applying the Kubelka–Munk function F(R):(1)$$F\left(R_{\infty }\right)=\frac{{\left(1-R_{\infty }\right)}^{2}}{{2R}_{\infty }}$$
where R_∞_ represents the measure of the diffuse reflectance. $F\left(R_{\infty }\right)$ could be utilized to estimate the material’s band gap energy ($E_{g}$) through Tauc’s method, which involved plotting ${\left[F\left(R_{\infty }\right)hv\right]}^{2}$ versus hv, where hv denotes the photon energy. By performing a linear fit to the slope of the curve and extrapolating this fit to its intersection with the horizontal axis, the band gap value was determined.

Figure 5a shows the Tauc plots for undoped and Ce-doped BaAl_2_O_4_ samples in an air atmosphere. These samples exhibited energy band gaps around 4 eV, consistent with values reported in the literature [23,24]. Figure 5b presents the Tauc plots for undoped and Ce-doped BaAl_2_O_4_ samples in a nitrogen atmosphere. In this case, two band gap values (4 eV, and 5 eV) were observed for the Ce-doped samples. However, undoped samples had a single energy value of 5.64 eV, in agreement with the findings of other authors [25]. The formation of multiple gaps of energy in the Ce-doped samples is attributed to the interaction between impurities and surface defects as suggested by Zhai et al. [26]. Thus, the air atmosphere favored a homogeneous structure, while the N_2_ atmosphere induced multiple energy states in the doped samples. The band gap values are in Table 2 for both sample groups.

The PL emissions for both series of samples, with an excitation wavelength of 300 nm, are presented in Figure 6. For the samples thermally treated in an oxidizing atmosphere, a broad emission band between 330 and 550 nm, centered around 357 nm, was observed for the doped samples, while the undoped sample exhibited a broad peak with a maximum at 391 nm. Pandey et al. [27] documented PL emission from undoped BaAl_2_O_4_ thermally treated in an oxidizing atmosphere, showing a broad emission band between 300 and 650 nm, with a maximum at 405 nm. Shivaramu et al. [25] presented the PL emission spectrum of undoped BaAl_2_O_4_ thermally treated in air, observing a narrow green emission peak at 542 nm and a broad emission band between 350 and 600 nm. Zhai et al. [28] reported the emission of undoped BaAl_2_O_4_ thermally treated in air, displaying a broad emission band from 350 to 550 nm with a maximum around 400 nm, which is consistent with the results presented here. Maphiri et al. [29] analyzed the PL emission of BaAl_2_O_4_ thermally treated in an oxidizing atmosphere, with two broad emission bands: the first between 400 and 650 nm and the second from 650 to 800 nm. Through deconvolution, maxima were identified around 420, 435, 457, 521, 612, and 722 nm. These emissions may have been due to defects in the crystalline structure, such as oxygen or barium vacancies. Additionally, Gedekar et al. [8] characterized the PL emissions of BaAl_2_O_4_: Ce^3+^ thermally treated in air, showing a broad band between 360 and 550 nm centered at 386 nm. These emissions may have been due to oxygen vacancies generated in the matrix by Ce^4+^ formation, some residual transitions from 4f^1^ to the 4f^0^5d^1^ state of Ce^3+^, and intrinsic structural defects in the matrix related to the material synthesis process. The results from this study are consistent with some of the findings reported in the literature, particularly regarding the broad emission bands observed. However, they differ from other reports. These differences may be attributed to variations in the synthesis conditions.

For the nitrogen-reducing case (Figure 6b), peaks centered around 386 nm were observed in the doped samples, with a broad peak centered at 422 nm in the undoped sample. Mao et al. [30] reported the PL emissions of undoped BaAl_2_O_4_ thermally treated in a reducing atmosphere (with carbon), observing a broad emission centered at 495 nm. Kumar et al. [3] investigated the PL emission of BaAl_2_O_4_ thermally treated in a reducing atmosphere and also with carbon, describing a broad band between 450 and 700 nm with a maximum centered at 589 nm. Chatterjee et al. [7] analyzed the PL emission of BaAl_2_O_4_ samples thermally treated in a reducing atmosphere with N_2_ and H_2_, showing a broad band between 370 and 515 nm with a maximum at 422 nm. These emissions were associated with intrinsic lattice defects and the 5d to ^2^F_5/2_ transitions of Ce^3+^. Our results are consistent with these latter findings. The differences in the PL emission maxima may be attributed to the distinct characteristics of the synthesis processes.

Figure 7 presents the integrated PL response as a function of Ce doping concentration for both sample series. In both cases, it was observed that the sample with 0.3% mol *Ce* exhibited the highest PL response. For the case of samples under an air atmosphere, the sample with 0.3 mol% *Ce* increased the PL response by 133% compared to the undoped sample, while for samples under a nitrogen atmosphere, the sample with 0.3 mol% Ce increased the PL response by 190% compared to the undoped sample. The use of a reduced atmosphere promoted higher emissions for this *Ce* concentration.

The chromaticity coordinates according to the International Commission on Illumination (CIE) for both prepared series were calculated and are shown in Figure 8. The (x, y) values were calculated using CIE chromaticity coordinate software, OriginPro version 2019b (OriginLab Corporation, Northampton, MA, USA) [31] and are presented in the tables included in the chromaticity graphs. The results indicate that the use of dopant influenced the blueish emission color in both sample series. The color purity can be described by the following equation [32]:(2)$$Color\;purity=\frac{\sqrt{{\left(x-x_{i}\right)}^{2}+{\left(y-y_{i}\right)}^{2}}}{\sqrt{{\left(x_{d}-x_{i}\right)}^{2}+{\left(y_{d}-y_{i}\right)}^{2}}}\times 100\%$$
where (x,y) are the chromaticity coordinates of the sample, $\left(x_{i},y_{i}\right)$ correspond to the white point (0.3101, 0.3162), and $\left(x_{d},y_{d}\right)$ are the coordinates of the dominant wavelength, in this case (0.1741, 0.005). Table 3 summarizes the calculated purities for BaAl_2_O_4_: x%Ce samples treated under air and nitrogen atmospheres.

These results indicate that the introduction of Ce significantly enhanced the color purity in both atmospheres, with the highest value (99.14%) observed at 0.5 mol% Ce in air. In this atmosphere, color purity increased with Ce concentration up to 0.5 mol%, after which it declined, likely due to concentration quenching, where non-radiative energy transfer between neighboring Ce ions became predominant.

In contrast, the samples treated in a nitrogen atmosphere exhibited more stable color purity values, ranging around 88.95% from 0.5 to 1.1 mol% Ce^3+^. This suggests that nitrogen may contribute to stabilizing the Ce^3+^ oxidation state and minimizing non-radiative losses, thus supporting more consistent emission properties.

The undoped samples displayed the lowest color purity values in both conditions, reinforcing the role of Ce^3+^ as an effective activator for enhancing luminescent performance. The differences between both atmospheres further suggest that the thermal treatment environment influenced defect formation and the stabilization of the Ce oxidation state.

Figure 9 presents the lifetime decay spectra for both series of samples under an excitation wavelength of 450 nm. These spectra were fitted using a biexponential function, as given by(3)$$y=y_{0}+A_{1}e^{-\frac{t}{\tau 1}}+A_{2}e^{-\frac{t}{\tau 2}}$$
where $A_{1}$ and $A_{2}$ were the weighting parameters and $\tau_{1}$ and $\tau_{2}$ were the components of the lifetime decay [3,32]. Using these parameters, the average lifetime decay could be calculated using the following equation:(4)$$\tau_{avg}=\frac{A_{1}\tau_{1}^{2}+A_{2}\tau_{2}^{2}}{A_{1}\tau_{1}+A_{2}\tau_{2}}$$

Table 4 and Table 5 list the parameters for each sample series, along with the calculated τ_avg_ values. The results showed that the thermal treatment in air and a N_2_ atmosphere impacted the average lifetime of the samples. The samples treated in air exhibited longer τ_avg_ values (0.94 ns for undoped BaAl_2_O_4_), which slightly decreased with increasing Ce concentrations (down to 0.79 ns for 1.1% Ce), suggesting that air treatment promoted radiative recombination. In contrast, the samples treated in a reduced atmosphere showed much shorter lifetimes (0.40 ns for undoped BaAl_2_O_4_ and around 0.34 ns for doped samples), indicating that oxygen reduction and the formation of structural defects, such as oxygen vacancies, accelerated non-radiative decay. This analysis suggests that the thermal treatment atmosphere played a crucial role in optimizing the luminescent properties of the material.

Kumar et al. [3] documented lifetimes for BaAl_2_O_4_: 0.1%Ce^3+^ with an approximate value of 3.42 ns. Hema et al. [18] measured a lifetime of 4 ns for BaAl_2_O_4_: 0.5%Ce^3+^. Maphiri et al. [29] identified a lifetime of BaAl_2_O_4_ with an approximate value of 503 ms. Chatterjee et al. [7] presented an average lifetime of BaAl_2_O_4_: 1%Ce^3+^ with a value of 8.3 µs. The latter lifetimes [7,29] were significantly longer than those obtained in this and other works [3,29]. This difference could be attributed to variations in synthesis methods, the specific Ce^3+^ concentration used, or thermal treatment conditions.

The observed decrease in lifetime values can be attributed to variations in experimental conditions that influence the photoluminescent properties of the materials. Structural defects, such as oxygen vacancies, may act as non-radiative recombination centers, while concentration quenching at higher Ce levels could promote non-radiative energy transfer processes. Additionally, the thermal treatment atmosphere affects both the oxidation state of cerium and the defect density, which in turn modifies relaxation mechanisms. These combined factors could explain the significant differences compared to other reported studies.

## 4. Conclusions

Two series of BaAl_2_O_4_ samples were prepared, with the cerium concentration varied from 0 to 1.1 mol%. The first series underwent heat treatment in an air atmosphere (Air Atm), while the second was treated in a nitrogen-reduced atmosphere (N_2_ Atm). XRD analysis confirmed the hexagonal crystal structure of BaAl_2_O_4_, and the bandgap values were calculated as 4 eV for air-treated samples and 5.6 eV for those treated in a nitrogen-reduced atmosphere. Broad PL emission bands were observed at approximately 357 nm and 386 nm for BaAl_2_O_4_: Ce samples treated in air and N_2_ atmospheres, respectively, with these emissions attributed to the dopant. XPS analysis confirmed that using a N_2_ atmosphere minimized cerium oxidation, thereby favoring the presence of trivalent Ce. Samples treated in air exhibited higher crystallinity, with larger crystallite sizes of 58.5 nm for undoped samples and 45.7 nm for doped ones. This suggests that the oxygen present in the atmosphere promoted the formation of a more ordered structure. Conversely, samples treated in a N_2_ atmosphere displayed smaller crystallite sizes, 39.8 nm for undoped samples and 42.3 nm for doped ones, likely indicating that oxygen reduction led to the formation of structural defects, such as oxygen vacancies. These defects hindered crystallite growth and reduced PL efficiency due to non-radiative processes, resulting in shorter lifetimes compared to those of the air-treated samples. Overall, our results indicate that the choice of environmental atmosphere for the thermal treatment significantly influences the structural and luminescent properties of BaAl_2_O_4_: Ce samples, making it a critical factor for the optimization of the luminescent properties of this material.

## Figures and Tables

**Figure 1 micromachines-16-00688-f001:**
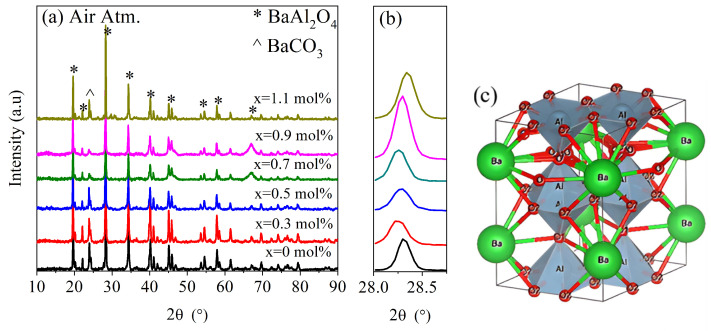
X-ray diffraction study for samples produced in air atmosphere. (**a**) Spectra of BaAl_2_O_4_: Ce samples, (**b**) magnification of (202) plane, and (**c**) hexagonal crystal structure of samples, where ions are indicated.

**Figure 2 micromachines-16-00688-f002:**
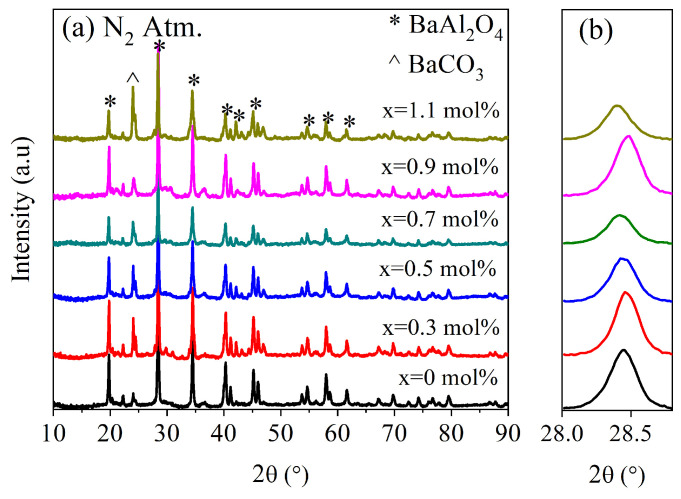
X-ray diffraction study for samples produced in N_2_ atmosphere. (**a**) Spectra of BaAl_2_O_4_: Ce samples. (**b**) Magnification of (202) plane, indicating variation in 2θ values with respect to Figure 1b.

**Figure 3 micromachines-16-00688-f003:**
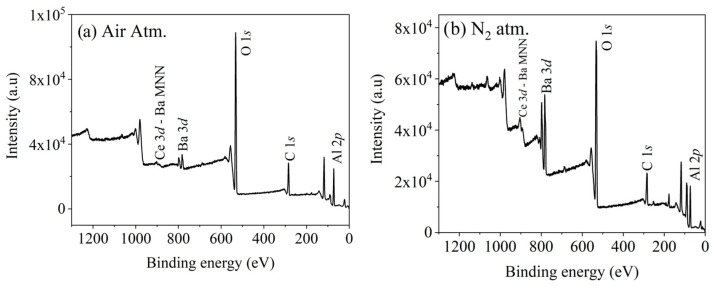
XPS spectra of BaAl_2_O_4_: Ce samples with 0.5 mol% Ce. (**a**) In air atmosphere. (**b**) In nitrogen-reduced atmosphere.

**Figure 4 micromachines-16-00688-f004:**
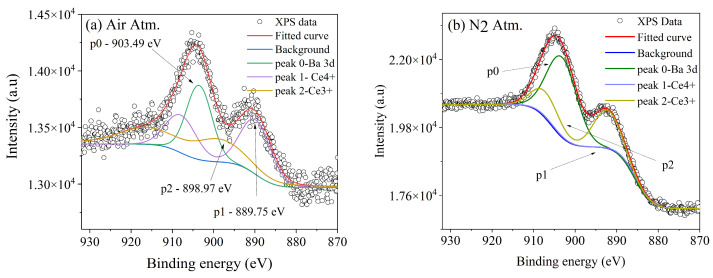
The deconvolution of the curve attributed to Ce *3d* for the BaAl_2_O_4_ samples with 0.5 mol% Ce (**a**) in an air atmosphere and (**b**) in a nitrogen atmosphere.

**Figure 5 micromachines-16-00688-f005:**
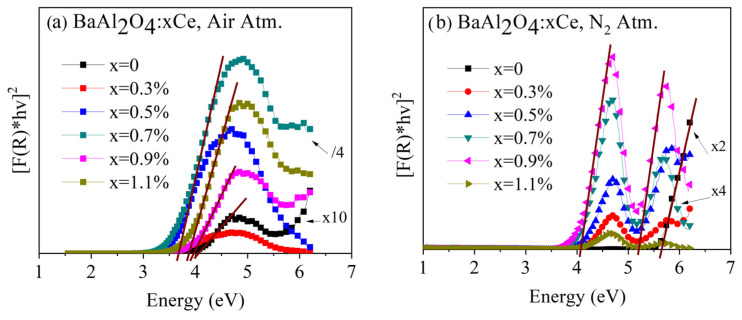
Diffuse reflectance spectra of BaAl_2_O_4_: Ce samples (**a**) heat-treated in air and (**b**) heat-treated in reduced atmosphere.

**Figure 6 micromachines-16-00688-f006:**
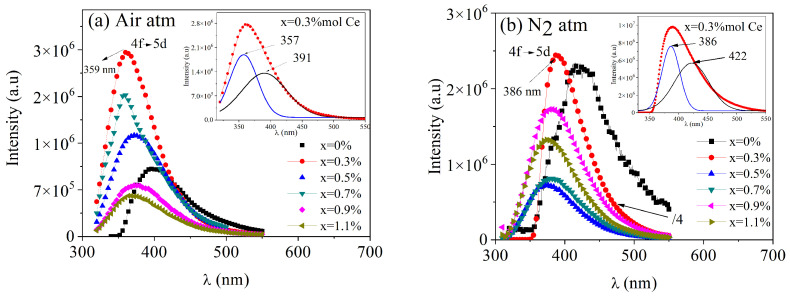
PL emission spectra of BaAl_2_O_4_: xCe samples (**a**) prepared in air and (**b**) in a nitrogen reduced atmosphere. Both samples were excited at a λ_ext_ = 300 nm wavelength.

**Figure 7 micromachines-16-00688-f007:**
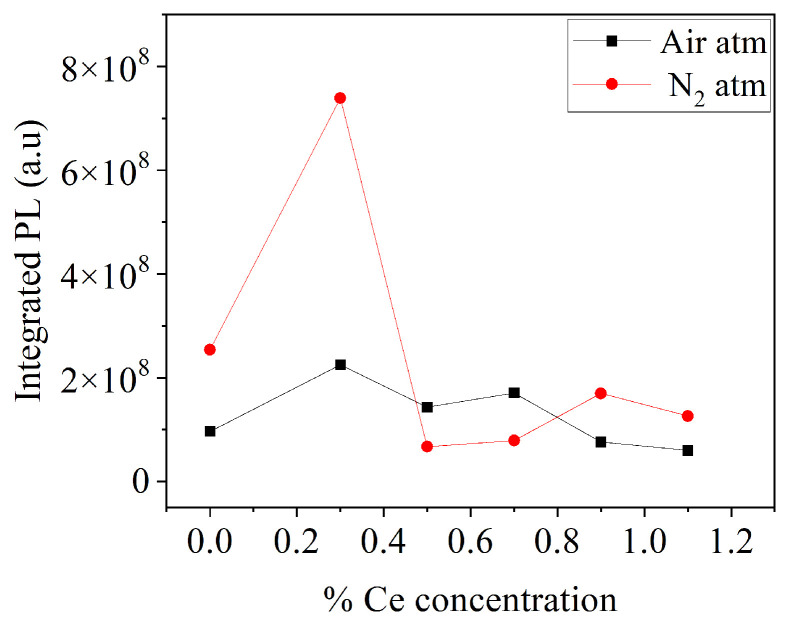
Integrated PL response of BaAl_2_O_4_ as function of Ce doping concentration.

**Figure 8 micromachines-16-00688-f008:**
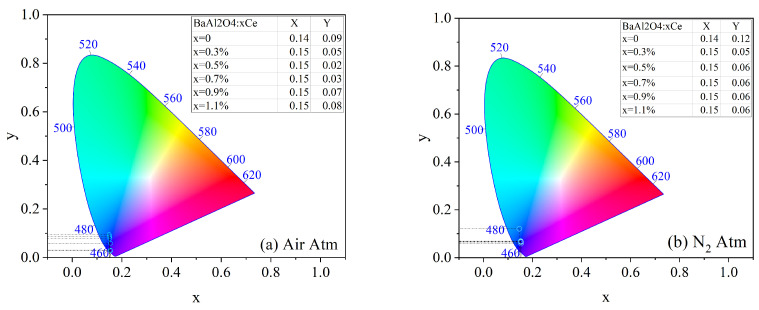
CIE coordinates of BaAl_2_O_4_: Ce samples with thermal treatment in (**a**) air atmosphere and (**b**) in reduced atmosphere.

**Figure 9 micromachines-16-00688-f009:**
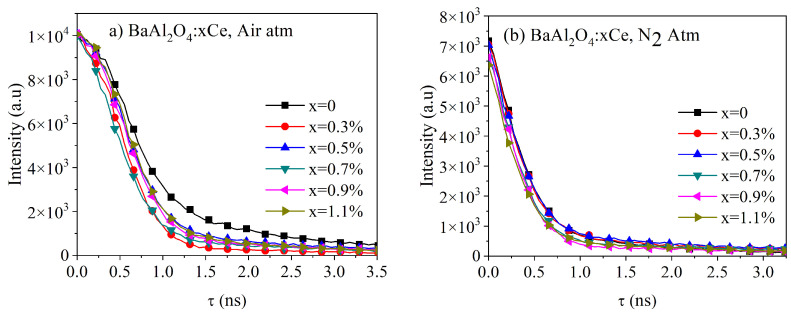
Luminescence decay spectra of BaAl_2_O_4_: Ce sample thermally treated in (**a**) air atmosphere and (**b**) N_2_ atmosphere. Both were excited at wavelength λ_ext_ = 450nm.

**Table 1 micromachines-16-00688-t001:** Crystal structure parameters of BaAl_2_O_4_: Ce samples in an oxidizing atmosphere (Air Atm) and in a reducing atmosphere (N_2_ Atm).

	Air Atm				N_2_ Atm			
[Ce] (mol%)	Crystallite Size (nm)	Lattice Parameters			Crystallite Size (nm)	Lattice Parameters		
		a (Å)	c (Å)	V (Å^3^)		a (Å)	c (Å)	V (Å^3^)
0	58.5	5.220	8.797	207.668	39.8	5.223	8.799	207.947
0.3	44.0	5.222	8.803	207.914	44.7	5.220	8.797	207.668
0.5	43.3	5.221	8.806	207.888	41.2	5.220	8.797	207.665
0.7	45.7	5.222	8.793	207.735	41.7	5.223	8.792	207.798
0.9	48.6	5.220	8.794	207.614	42.7	5.225	8.799	208.073
1.1	46.9	5.221	8.795	207.653	41.2	5.223	8.796	207.842

**Table 2 micromachines-16-00688-t002:** Band gap values of BaAl_2_O_4_ and BaAl_2_O_4_: Ce samples prepared under different atmospheres.

Air Atm		$N_{2}$ Atm		
%Ce	Eg Value (eV)	%Ce	Eg1 (eV)	Eg2 (eV)
0	4	0	-	5.64
0.3	3.58	0.3	4.16	5.14
0.5	3.54	0.5	4.1	5.25
0.7	3.57	0.7	4.04	5.17
0.9	3.97	0.9	4.06	5.19
1.1	3.87	1.1	4.06	5.19

**Table 3 micromachines-16-00688-t003:** Chromaticity coordinates and color purity of Ce^3+^-doped BaAl_2_O_4_ under air and nitrogen atmospheres.

Atmosphere	% Ce	(x, y) Coordinates	Color Purity (%)
Air	0.0	(0.14, 0.09)	83.33
Air	0.3	(0.15, 0.05)	91.46
Air	0.5	(0.15, 0.02)	99.14
Air	0.7	(0.15, 0.03)	96.56
Air	0.9	(0.15, 0.07)	86.47
Air	1.1	(0.15, 0.08)	84.01
N_2_	0.0	(0.14, 0.12)	76.45
N_2_	0.3	(0.15, 0.05)	91.46
N_2_	0.5–1.1	(0.15, 0.06)	88.95

**Table 4 micromachines-16-00688-t004:** Fit parameters (A_1_, A_2_) for average lifetime (τ_av_) calculation for BaAl_2_O_4_: Ce samples thermally treated in Air Atm.

Sample	$A_{1}$	$A_{2}$	$\tau_{1}$ (ns)	$\tau_{2}$ (ns)	$\tau_{avg}$ (ns)
$BaAl_{2}O_{4}$	2.07755 × 10^42^	2.07755 × 10^42^	0.94	0.94	0.94
$BaAl_{2}O_{4}:0.3\%Ce$	2.23482 × 10^49^	2.23482 × 10^49^	0.80	0.80	0.80
$BaAl_{2}O_{4}:0.5\%Ce$	2.31252 × 10^55^	2.31252 × 10^55^	0.70	0.70	0.70
$BaAl_{2}O_{4}:0.7\%Ce$	4.35158 × 10^52^	4.35158 × 10^52^	0.74	0.74	0.74
$BaAl_{2}O_{4}:0.9\%Ce$	6.78123 × 10^50^	6.78123 × 10^50^	0.77	0.77	0.77
$BaAl_{2}O_{4}:1.1\%Ce$	6.82335 × 10^49^	6.82335 × 10^49^	0.79	0.79	0.79

**Table 5 micromachines-16-00688-t005:** Fit parameters (A_1_, A_2_) for average lifetime (τ_av_) calculation for BaAl_2_O_4_: Ce samples thermally treated in N_2_ Atm.

Sample	$A_{1}$	$A_{2}$	$\tau_{1}$ (ns)	$\tau_{2}$ (ns)	$\tau_{avg}$ (ns)
$BaAl_{2}O_{4}$	3737.23	3737.23	0.40	0.40	0.40
$BaAl_{2}O_{4}:0.3\%Ce$	3655.90	3655.90	0.38	0.38	0.38
$BaAl_{2}O_{4}:0.5\%Ce$	3550.57	3550.57	0.39	0.39	0.39
$BaAl_{2}O_{4}:0.7\%Ce$	3445.15	3445.15	0.35	0.35	0.35
$BaAl_{2}O_{4}:0.9\%Ce$	3498.45	3498.41	0.34	0.34	0.34
$BaAl_{2}O_{4}:1.1\%Ce$	3248.03	3248.03	0.34	0.34	0.34

## Data Availability

Data are contained within the article; nevertheless, if more detailed information is required, the dataset will be available on request from the authors.

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
