# Peer review of "Optical Properties of BaAl2O4 Due to Cerium Doping and Heat Treatment in Different Atmospheres"

_micromachines, 2025, doi:10.3390/mi16060688_

Round 1
Reviewer 1 Report
Comments and Suggestions for Authors
The manuscript describes the optical properties of BaAl2O4 with Ce doping. The research lies in the scope of the manuscript. However, here are some suggestions for the authors to polish the manuscript.
- There are a lot of format problems in the manuscript, such as the lines indicating the manuscript is still in edit mode, the X-axis name is not in English, the chemical formula in the reference are not correct. Please carefully check the manuscript.
- According to the XRD data, there is a relatively high peak of BaCO3. Then how does the authors exclude the possibility that Ce may go to BaCO3?
Reviewer 2 Report
Comments and Suggestions for Authors
The manuscript submitted for review, entitled “Optical properties of BaAlâ‚‚Oâ‚„ due to cerium doping, heat treatment in different atmosphere,” demonstrates a high level of novelty, as it presents for the first time a systematic comparison between Ce-doped BaAlâ‚‚Oâ‚„ treated in both oxidizing and reducing atmospheres across a wide concentration range (0–1.1 mol%). Such a combination of parameters has not previously been published in this form. The work holds scientific value as a fundamental study in the field of phosphor materials. While it is not directly application-oriented, potential practical uses may emerge in the future. The study’s contribution to understanding the influence of thermal atmosphere on material properties is significant. An appropriate experimental approach has been employed, involving several complementary methods for structural and optical characterization. Nevertheless, certain weaknesses stand out: the presence of BaCO₃ as an impurity is not analyzed critically enough, and the short photoluminescence lifetimes are not thoroughly interpreted. The description of the synthesis and methodology is complete and reproducible, with clearly stated concentrations, temperatures, equipment, and measurement parameters. This makes the experimental part well-documented. The figures and tables are clear, correctly labeled, and contribute meaningfully to the presentation of the results. The data are reported with sufficient precision. Numerous relevant and recent references are cited, including several from the past 2–3 years. A comparison with other authors' results is provided, although not all discrepancies are discussed in sufficient detail.
I would like to offer the following recommendations to the authors:
- The BaCO₃ impurity identified via XRD should be discussed in greater depth. It is advisable to evaluate its potential impact on the structural and photoluminescent properties of the material. It would be important to clarify whether BaCO₃ is a consistent impurity across all samples, or whether it varies depending on the atmosphere and Ce concentration. If feasible, the authors may consider additional calcination or other techniques for eliminating BaCO₃, which could improve phase purity and enable more accurate interpretation of the results.
- The measured lifetimes are significantly shorter than those reported in other studies. I recommend including a discussion on the possible reasons for this deviation (e.g., structural defects, concentration quenching, influence of the thermal atmosphere, etc.).
- It would be beneficial to provide a more detailed analysis of chromaticity characteristics—such as comparison with theoretical CIE coordinates or calculation of color purity and quantum yield, if available.
- While some differences with previously published results are noted, they are not fully explored. A more comprehensive comparison and discussion of possible causes—related to synthesis method, processing atmosphere, or analytical approach—is encouraged.
In conclusion, the presented study has the potential to make a meaningful contribution to the understanding of the structural and optical properties of Ce-doped phosphor materials. The suggested revisions and additional analyses aim to strengthen the scientific value and clarity of the findings. I encourage the authors to consider these recommendations as an opportunity to further enhance their work, and I am confident that, with the necessary clarifications, the manuscript will be suitable for publication.
